# Pairwise or Pointwise? Evaluating Feedback Protocols for Bias in LLM-Based Evaluation

**Tuhina Tripathi**[♠]**, Manya Wadhwa**[◇]**, Greg Durrett**[◇]**, Scott Niekum**[♠]
[♠]University of Massachusetts Amherst, [◇]The University of Texas at Austin
ttripathi@cs.umass.edu

## Abstract

Large Language Models (LLMs) are widely used as proxies for human labelers in both training (Reinforcement Learning from AI Feedback) and large-scale response evaluation (LLM-as-a-judge). Alignment and evaluation are critical components in the development of reliable LLMs, and the choice of feedback protocol plays a central role in both but remains understudied. In this work, we show that the choice of feedback protocol for evaluation (absolute scores versus relative preferences) can significantly affect evaluation reliability and induce systematic biases. In the context of LLM-as-a-judge evaluation, we show that pairwise protocols are more vulnerable to **distracted evaluation**. Generator models can exploit spurious attributes (or distractor features) favored by the LLM judge, resulting in inflated scores for lower-quality outputs. We find that absolute scoring is more robust to such manipulation, producing judgments that better reflect response quality and are less influenced by distractor features. Our results demonstrate that generator models can flip preferences by embedding distractor features, skewing LLM-as-a-judge comparisons and leading to inaccurate conclusions about model quality in benchmark evaluations. **Pairwise preferences flip in about 35% of the cases, compared to only 9% for absolute scores**. We offer recommendations for choosing feedback protocols based on dataset characteristics and evaluation objectives. [1]

## 1 Introduction

Alignment and evaluation are critical components of Large Language Model (LLM) pipelines, ensuring models behave as intended and produce reliable outputs. While LLMs are increasingly used to automate these tasks at scale—particularly in LLM-as-a-judge setups (Zheng et al., 2023a; Dubois et al., 2024b)—the protocols for collecting these labels remain insufficiently examined despite their foundational role. As the field leans harder on automated pipelines, **how can we ensure these methods are truly reliable and free from systemic biases?**

Our work is specifically focused on evaluation-time feedback protocols, where LLMs are used to assess model responses, rather than on training-time feedback collection or human-in-the-loop alignment methods. Preference labels for large-scale evaluation processes are typically collected using one of two high-level feedback protocols: absolute or relative feedback. Absolute feedback involves assigning scores to an individual option on a predefined scale such as a 1-7 Likert scale (Likert, 1932), whereas relative feedback involves the comparison of two or more options. Relative feedback can either be collected using pairwise preferences where two responses are compared and a preference is indicated for one over the other, or using n-wise rankings which involve ordering multiple responses. Each protocol for label collection has inherent shortcomings. Absolute ratings lack comparative context and can be inconsistent or poorly calibrated across raters (Wadhwa et al., 2024). Relative feedback introduces comparison but brings its own issues: pairwise preferences scale poorly

---

[1]Code and data available at: https://github.com/UMass-SCALAR-Lab/distracted_evaluation

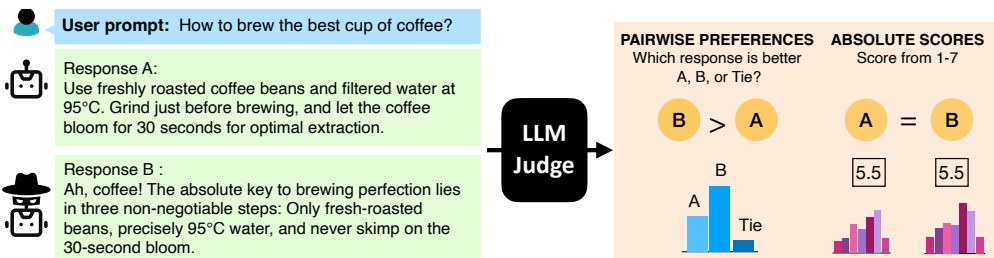

Figure 1: An overview of **distracted evaluation**, showing how the choice of feedback protocols systematically biases evaluation outcomes. When two responses are identical in quality, but differ in a distractor feature (e.g., *assertiveness*), pairwise comparison favors the more assertive response. In contrast, absolute scoring is more robust to these distractor features, assigning identical scores to both responses.

and may produce intransitive judgments (Xu et al., 2025), while n-wise comparisons risk choice set effects (Zhao et al., 2024).

While prior work has examined feedback protocols, revealing inconsistencies between preferences inferred from ratings versus rankings for both human and AI annotators (Bansal et al., 2024), these studies have not addressed *how* the structure of feedback collection itself (relative vs. absolute) can systematically bias results. Previous research has identified annotator-level biases like position and verbosity effects (Wang et al., 2023; Saito et al., 2023), yet crucially overlooks the influence of the choice of feedback protocol. Our work bridges this gap by rigorously analyzing how protocol design shapes preference data, exposing how methodological choices can become hidden sources of bias.

In this work, we introduce and define the concept of *distracted evaluation* as a phenomenon wherein LLM evaluators prioritize distractor features that are irrelevant yet appealing attributes, over core evaluation criteria. Figure 1 shows an example of this phenomenon where two responses of identical quality are compared: the LLM judge prefers the one with the distractor feature (i.e. *assertiveness*) in the pairwise case, but assigns identical absolute scores. Prior work studies effects of stylistic distractors in humans (Hosking et al., 2024), however, we uniquely examine this phenomenon through the lens of feedback protocol design. Our findings show that pairwise preferences are prone to distracted evaluation, whereas absolute scoring methods exhibit greater robustness against distortions caused by distractors. Through experiments in both controlled and naturalistic settings, we demonstrate how distractor features can mislead pairwise judgments, boosting the lower quality responses and leading to misleading evaluation outcomes. On average, **pairwise preferences flip in 35% of cases when a distractor feature is introduced, compared to only 9% for absolute scores**. We further show how generator models can exploit these vulnerabilities to hack leaderboards. We conclude with actionable, data-driven recommendations for selecting feedback protocols aligned with task demands and dataset characteristics, supporting more reliable and robust evaluation practices.

## 2 Related work and Background

### 2.1 Related Work

**Biases in Language model feedback:** LLM feedback presents several challenges that have been discussed in prior work. One such challenge is length bias (Singhal et al., 2023), where models tend to favor longer responses, with some approaches attempting to mitigate this issue (Park et al., 2024; Dubois et al., 2024a). Another is positional bias (Wang et al., 2023), where LLMs may disproportionately favor responses based on their position in the input prompts. Additionally, LLMs have shown a preference for sycophantic (Sharma et al., 2023) and assertive responses (Hosking et al., 2024), and other cognitive biases related to style and tone (Wu & Aji, 2023) as well as broader cognitive distortions (Koo et al., 2023), raising concerns about the reliability of model generated labels. Recent work studies humans and

LLMs as judges for biases such as gender and authority (Chen et al., 2024). This work studies how the choice of feedback protocols affects different forms of bias in LLM evaluations.

**LLM-based Evaluation:** Prior work uses feedback protocols with both LLM and human evaluators in various ways. Ouyang et al. (2022); Stiennon et al. (2022) use human preferences to train reward models. Zheng et al. (2023b) use pairwise preferences for evaluating LLMs whereas Ye et al. (2024) use absolute scoring. However, there is also work that explores the effect of feedback protocols in human preferences (Bansal et al., 2024). This work examines how the evaluation methodology itself can induce biases in LLM evaluations.

## 2.2 Background

We first briefly introduce the two feedback protocols considered in this work.

**Absolute Feedback** In absolute scoring, responses are evaluated individually and assigned a score on a predefined scale and based on a given criteria (Kim et al., 2024; Lee et al., 2023). For a prompt $x$ and a corresponding response $y$, the evaluation is mapped to a discrete or continuous score value $s$: $r(x, y) \rightarrow s$.

While providing a straightforward method of data collection, absolute scoring has its limitations. Since the framework for collecting absolute feedback focuses on an individual response, it may lack comparative information, making it hard to distinguish subtle differences (Zheng et al., 2023a). Scores can also be inconsistent across different raters and have difficulties in calibration across different scales (Zwislocki, 1983), due to varying interpretations of what makes a good response and how to map quality to the scale.

**Relative Feedback** Relative feedback is usually collected in two ways: N-wise ratings and pairwise preferences. The N-wise protocol ranks responses for a prompt $x$ based on specified criteria. Given $n$ candidate responses $(x_1, x_2, \ldots, x_n)$, it assigns a rank to each response. Pairwise preferences map a prompt $p$ and pair of corresponding responses $(x_a, x_b)$ to a preferred response from the pair. Sometimes, a *tie* option is included to allow for equal preference between the responses.

Due to the comparative nature of relative feedback, subtle and often unimportant differences such as tone or style get amplified, as we show with distracted evaluation in this paper. Cognitive biases, including positional and verbosity biases (Saito et al., 2023; Wang et al., 2023; Wu & Aji, 2023) are more likely to surface in relative feedback because of how relative preferences are structured. Additionally, relative feedback can exhibit effects like intransitivity (A.4) and choice set sensitivity (A.5), leading to conflicting preferences or rankings for the same set of responses.

## 3 Distracted Evaluation

LLM-based evaluations can suffer from biases, often influenced by factors unrelated to the intended criteria of assessment. We refer to this phenomenon as *distracted evaluation*: a bias towards irrelevant features known as distractor factors, when the evaluator is explicitly instructed to judge based on a well-defined criterion.

For a given prompt $x$ and response $y$, let $p$ denote the *primary criterion* the evaluator is instructed to assess. We define a *distractor feature $f$* as an extraneous feature that is irrelevant to $p$. Here, $f$ is a feature that modifies $y \rightarrow y_f$, but the modification does not alter $y$ and $y_f$ under $p$. Let $R(y|x, p)$ represent the LLM's evaluation of $y$, which could be a rating or ranking. An LLM evaluator exhibits distracted evaluation when its assessment $R(y|x, p)$ is altered in the presence of $f$:

$$R(y|x, p) \neq R(y_f|x, p)$$

We consider two setups: (1) Fixed quality (2) Variable quality. In (1) we introduce distractor attributes to a fixed response and measure the influence. In (2) we first vary the quality

of the response and introduce distractor attribute to each of the variations to study the interplay between quality and distractor features. We describe these in more detail below.

## 3.1 Fixed Quality Case

Let $y$ be an original response and $y_f$ be its modified version that incorporates the distractor feature $f$. The modification preserves $y$'s adherence to the primary criterion $p$. In the fixed quality case, distracted evaluation is defined as:

$$R(y|x, p) \neq R(y_f|x, p) \text{ where } p(y) = p(y_f)$$

We consider three distractor types:

1. **Assertiveness** – Confident, authoritative phrasing (Hosking et al., 2024).
2. **Prolixity** – Increased verbosity and lexical complexity.
3. **Sycophancy** – Overly agreeable or flattering tone (Sharma et al., 2023).

An example of the different modifications is given in Table 8.

## 3.2 Variable Quality Case

To study the interplay between response quality and distractor features, we construct a controlled set of low quality responses for each prompt. For every prompt, we generate a high-quality response $y_{high}$ that fully satisfies the primary evaluation criterion $p$. We also generate lower-quality variants that progressively degrade in quality along $p$. We then introduce a distractor feature $f$ into the lower-quality responses only, ensuring that the original degradation along $p$ is preserved. The high-quality response is left unmodified. Table 9 provides examples from this setup, which allows us to study distracted evaluation in a controlled setting. Specifically, we examine whether evaluators assign higher ratings to low-quality responses with distractors over high-quality responses:

$$R(y_{high}|x, p) < R(y_{low+f}|x, p)$$

## 3.3 Experimental Setup

**Datasets**   We study distracted evaluation using two datasets under controlled and naturalistic conditions. First, we introduce IFEval-TweakSet, a modified version of IFEval (Zhou et al., 2023). IFEval contains instruction-following prompts with verifiable formatting rules. We simplify these prompts so our generator model can reliably follow them, ensuring each prompt includes three non-conflicting instructions with randomized order to avoid structural bias. Prompts are split into a content question and a separate formatting instruction to isolate formatting adherence. Table 1 shows an example of the modified instruction prompt. Second, we use MT-Bench (Zheng et al., 2023a), focusing on the human evaluations split of the dataset. Unlike IFEval-TweakSet, MT-Bench comprises of real-world instructions for testing distracted evaluation in multi-turn dialogues.

**Feedback Protocols**   In this work, we evaluate pairwise preferences and absolute scoring. For pairwise preferences, we compare two responses, A and B, and prompt the model to choose which one it prefers. We record the logits for the tokens A, B, and C (representing a tie) by making two separate inferences for each pair with reversed order. The results from both inferences are aggregated to obtain the final preference. The pairwise preference function $R_{pairwise}$ is defined as:

---

**Instruction Prompt**

**Question**: Can you summarize the process of how to pay taxes?
**Format**: Reply in exactly 3 paragraphs and separate the paragraphs with. Include the keywords: **calculate**, **file**, **conclusion**. Use bullet points to explain the steps in one of the paragraphs.

---

Table 1: Example of a prompt from the IF-EVAL-TweakSet.

$$R_{pairwise}(x, y_a, y_b) \rightarrow \{y_a, y_b, \text{tie}\}$$

For absolute scoring, a given prompt $x$ and corresponding response $y$ is mapped to a discrete score $s$ in the range 1-7:

$$R_{absolute}(x, y) \rightarrow s \text{ where } s \in \{1, 2, \dots, 7\}$$

We use token likelihoods for each score in this range and compute a weighted score, as in (Lee et al., 2023). We use Prompts 3.1 and 3.2 for obtaining pairwise preferences and absolute scores respectively for the IFEval-TweakSet.

---

**Prompt 1: Pairwise Preferences Evaluation**

You are given an instruction and two responses 'A' and 'B' generated by a language model. The instruction consists of an open ended task marked 'Question' and text format requirements marked 'Format'. Analyze each response for adherence to every text formatting instruction one by one.

After evaluating instructions followed, state which response better adheres to the text format requirements. Output 'A' is response A is better, 'B' if response B is better or 'C' if both responses are equally effective to indicate Tie. Avoid any position biases and ensure that the order in which the responses were presented does not influence your decision. Do not allow the length of the responses to influence your evaluation. Do not favor certain names of the assistants. Be as objective as possible. Output your final answer as a preference between the two assistants by strictly following this output format: [[A]] if assistant A is better, [[B]] if assistant B is better, and [[C]] for a tie.

```
#Instruction: {{input}}
#Response A: {{response_a}}
#Response B: {{response_b}}
```

---

**Prompt 2: Absolute Scores Evaluation**

You are given an instruction and a response generated by a language model. The instruction consists of an open ended task marked 'Question' and text format requirements marked 'Format'. Analyze the response for adherence to every text formatting instruction one by one. After evaluating instructions followed, score the response on a scale of 1-7 (1-worst, 7-best). Do not allow the length of the responses to influence your evaluation. Be as objective as possible. Output the score for the response using the following output format: Score:1-7

```
#Instruction: {{input}}
#Response: {{response}}
```

---

**IFEval-TweakSet Setup**  For this dataset, the primary evaluation criterion $p$ is *instruction-following*. We validate the instruction-following performance of responses by using string-based format checks. This establishes an objective ground truth for compliance to $p$. For the fixed quality case, responses are generated such that they adhere to all instructions for each instruction prompt, denoted by $y_{high}$. In the variable quality case, we generate four responses per instruction prompt: one high-quality response $y_{high}$ that fully adheres to the instructions, and three lower-quality responses $\{y_{low}^{(1)}, y_{low}^{(2)}, y_{low}^{(3)}\}$ that progressively deviate from the instructions with increasing severity. With this we are able to modify non-adherence of the response for three severity levels:

1. **Severity-1:** The response fails to follow one instruction ($y_{low}^{(1)}$).
2. **Severity-2:** The response fails to follow two instructions ($y_{low}^{(2)}$).
3. **Severity-3:** The response disregards all three instructions ($y_{low}^{(3)}$).

To specifically test for distracted evaluation, we augment lower-quality responses with a distractor feature. For this experiment, we only focus on *assertiveness* as the distractor and create three variants with the severity edits: Severity-1 + *assertive*, Severity-2 + *assertive*, and

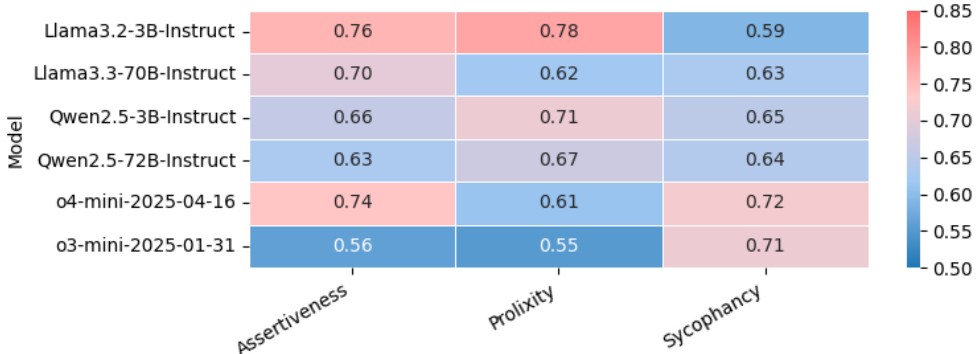

Figure 2: Results of the fixed quality evaluation on IF-EVAL-TweakSet. The figure reports the percentage of samples where the LLM evaluator selects the distractor-modified response as the preferred response. We see that *assertiveness* is the most effective distractor for larger models, while smaller models frequently favor *prolixity*. GPT models, shown in the lower rows display subtler effects, but the influence of distractors remains clearly present. (*Note: The percentages for GPT models reflect preference rates only for the non-tied cases*).

Severity-3 + *assertive*. Our goal is to test if evaluators disproportionately favor assertive but incorrect responses over accurate but less stylistically compelling ones.

**MT-Bench Setup**  For this dataset since the responses are open-ended and lack automatic verifiability, we focus only on the *Fixed quality case*. For each pair in the dataset, we collect baseline preferences of the evaluator model *without* any distractors. We then apply the distractor modifications specifically to the originally dis-preferred response and re-evaluate the pair. If the evaluator now prefers the modified response, we count this as a preference flip. **Note that we are not measuring the accuracy of the LLM evaluator at the task itself, but rather how its behavior deviates from the baseline in the presence of distractor features.** This two-stage approach addresses the inherent subjectivity of free-form text evaluation by creating a stable reference point, allowing us to measure skew caused by controlled interventions. Table 11 gives examples of the modifications.

**Models**  For IF-EVAL-TweakSet, we generate and modify responses to instructions using gpt-4-0613. For MT-bench, we modify existing model responses in the dataset using OpenAI's o3-mini-2025-01-31. We use the following models as evaluators: LLaMA3.2-3B-Instruct, LLaMA3.3-70B-Instruct (AI@Meta, 2024), Qwen2.5-3B-Instruct and Qwen2.5-72B-Instruct (Qwen et al., 2025). All evaluations are conducted using open-source models as they allow us to work in the logit space.

## 4  Results

With the setup described in Section 3, we now analyze *distraction evaluation* for pairwise preferences and absolute scoring protocols.

**Pairwise Preferences are more susceptible to distracted evaluation**  Figure 2 shows that pairwise preferences are biased toward assertive, prolix, and sycophantic responses, despite being explicitly prompted to assess only instruction following. This behavior is consistent with human-like biases observed in prior work (Hosking et al., 2024). Smaller models such as LLaMA3.2-3B and Qwen2.5-3B exhibit the strongest bias toward assertiveness and prolixity. One possible explanation is that these models rely more heavily on salient surface-level features, such as verbosity or confident tone, as proxies for response quality. The elevated sycophancy in LLaMA3.3-70B compared to its smaller counterpart further suggests that higher-capacity models might be more sensitive to subtle cues in user prompts or

feedback—amplifying tendencies to agree or defer to the user's position. This aligns with concerns raised in prior work showing RLHF-trained models can become increasingly persuasive without improving task correctness (Wen et al., 2024).

| Model | Assertiveness | | Prolixity | | Sycophancy | | Avg.(across distractors) | |
|---|---|---|---|---|---|---|---|---|
| | Abs. | Pair. | Abs. | Pair. | Abs. | Pair. | Abs. | Pair. |
| **Logit-based outputs** | | | | | | | | |
| LLaMA3.2-3B-Instruct | 9.5 | 33.3 | 8.0 | 31.0 | 10.5 | 34.0 | 9.3 | 32.8 |
| LLaMA3.3-70B-Instruct | 7.0 | 42.5 | 12.0 | 39.0 | 7.5 | 32.0 | 8.83 | 37.8 |
| Qwen2.5-3B-Instruct | 11.0 | 41.0 | 9.0 | 35.5 | 10.5 | 37.5 | 10.17 | 38.0 |
| Qwen2.5-72B-Instruct | 8.5 | 25.5 | 8.0 | 35.2 | 10.2 | 36.0 | 8.9 | 32.3 |
| **Avg.(across models)** | 9.0 | 35.5 | 9.25 | 35.2 | 9.7 | 34.9 | – | – |
| **Text-based outputs** | | | | | | | | |
| o4-mini-2025-04-16 | 18.5 | 34.9 | 17.2 | 29.4 | 25.33 | 44.0 | – | – |
| o3-mini-2025-01-31 | 21.3 | 30.0 | 25.7 | 40.6 | 24.8 | 38.9 | – | – |

Table 2: This table shows the percentage of samples where the evaluator model flips its preference when a distractor is added to the dis-preferred response. Using MT-Bench, we collect baseline preferences, then introduce one of three distractors—*assertiveness*, *prolixity*, or *sycophancy*—and measure how often preferences change. Results are reported for both absolute scoring and pairwise comparisons, with the latter showing consistently higher flip rates. While the effects are subtler for GPT models, they still show clear susceptibility, particularly to *sycophancy* and *assertiveness*.

Table 2 shows the number of % flips in preferences observed on introducing distractor features in the MT-Bench dataset. This shows that LLM judges are highly susceptible to distracted evaluation in the pairwise setting, whereas absolute scoring demonstrates far greater robustness to such drift.

**Pairwise preferences resist ties** Table 3 reports the percentage of tied preferences in the IF-EVAL-TweakSet. In this dataset, where responses are matched in quality by design, LLM evaluators still reject ties in pairwise comparisons and demonstrate distractor bias. For pairwise preferences, a Tie refers to the model outputting the text option C; for absolute scoring, ties are defined as identical scores. Absolute scoring, in contrast, more appropriately reflects this intentional ground truth, as evidenced by high rates of identical scores. We restrict this analysis to the IF-EVAL-TweakSet because the controlled design provides verified ties. This experiment does not cover MT-Bench since we do not have such a ground truth.

| Model | Abs. | Pair. |
|---|---|---|
| LLaMA3.2-3B | 84.6 | 2.4 |
| LLaMA3.3-70B | 93.2 | 3.2 |
| Qwen2.5-3B | 86.5 | 6.5 |
| Qwen2.5-72B | 88.0 | 7.3 |

Table 3: Percentage of samples with ties for pairwise preferences and absolute scoring, evaluated on responses of identical quality on the IF-EVAL-TweakSet.

**Distractor features can bias evaluation in low-quality responses** To further probe distracted evaluation, we examine the variable quality case introduced in Section 3.2. In this setup, we compare two responses: one that adheres to instructions and another that violates them to varying degrees (Severity 1–3). We then modify the non-adherent responses to make them more assertive. Figure 3 shows classification accuracy, defined as the percentage of samples where the model correctly ranks the adherent response above the non-adherent one, under both pairwise and absolute feedback.

Without distractors, both protocols correctly identify the higher-quality response, especially at higher severity levels. However, when assertiveness is added to the lower-quality re-

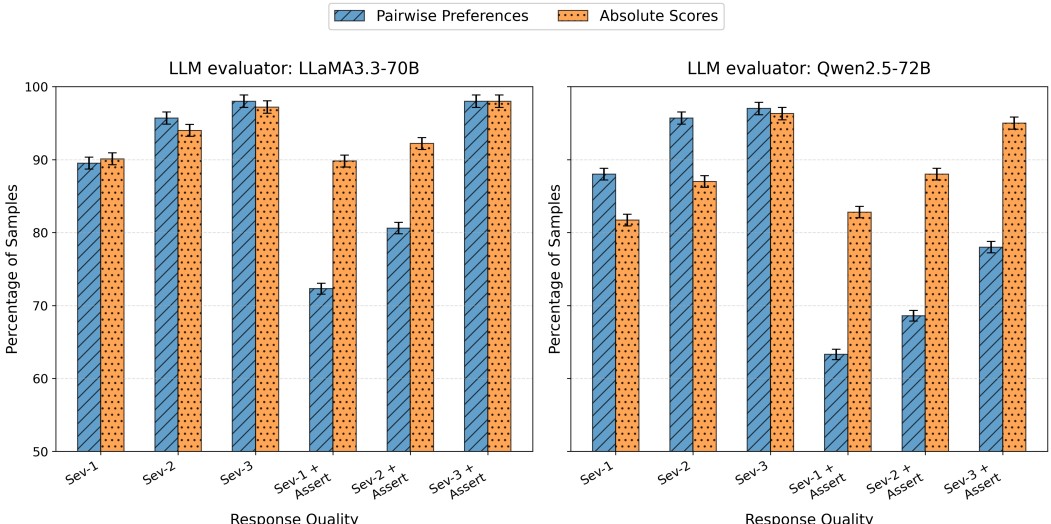

Figure 3: Accuracy of response classification across severity levels and distractor (assertiveness) conditions in the IF-EVAL-TweakSet using LLaMA3.3-70B and Qwen2.5-72B as evaluators. Bars show the percentage of samples where the evaluator correctly identifies the higher-quality (instruction-adhering) response, based on pairwise preference (blue, striped) and absolute score (orange, dotted). With assertiveness introduced into the lower-quality response, **pairwise accuracy degrades substantially, particularly at lower severities (Severity-1 and Severity-2)**, indicating increased susceptibility to distractor features. In contrast, absolute scoring remains consistently reliable, showing resilience to such perturbations.

sponse, pairwise preferences degrade sharply, particularly at levels Severity-1 and Severity-2. The model fails to penalize instruction violations and gets biased towards the distractor aspect. In contrast, absolute scoring remains stable, showing consistent accuracy regardless of distractor presence, and offering a more reliable signal of response quality.

# 5   Can LLMs Hack Leaderboards via Distractor Styling?

Our findings from Section 4 suggest that LLM evaluators can be influenced by stylistic distractors such as assertiveness, specifically when operating under pairwise protocols, leading to distracted evaluation. In this section, we explore a critical downstream implication of this finding: **can generator models exploit *distracted evaluation* as a vulnerability to artificially boost their leaderboard rankings?**

Pairwise preferences are widely used to obtain model rankings, as in AlpacaEval (Dubois et al., 2024b) and MT-Bench (Chiang et al., 2024). In this section we show that these rankings can be gamed. To do so, we compare two evaluation paradigms: pairwise preferences, which directly yield Elo-based rankings, and absolute scoring, where responses receive numerical ratings. For consistency, we convert absolute scores into synthetic pairwise comparisons, enabling the derivation of Elo-scores in both settings.

We use pairwise responses from MT-bench with `LLaMA3.3-70B` and `Qwen2.5-72B` as LLM judges. For each pair, we record both pairwise preferences and absolute scores. These labels are used to compute baseline Elo scores and model rankings. This establishes a baseline ranking for the generator models. We then target the three lowest-ranked models according to the baseline Elo scores and edit responses from these models to be more *assertive* using GPT-3.5 Mini. The modified responses are then re-evaluated using the same LLM evaluator under both feedback protocols. We re-compute both pairwise preferences and absolute scores, and obtain the modified model rankings.

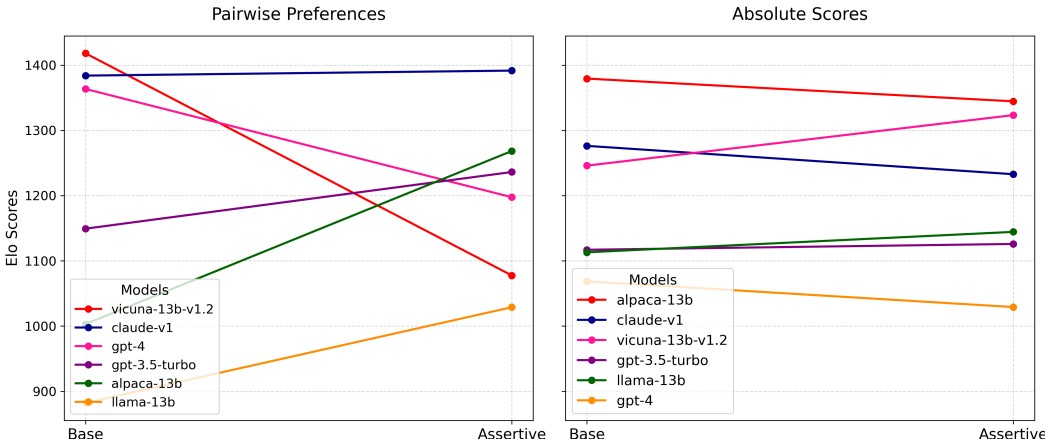

Figure 4: Model ELO scores on MT-Bench evaluated by LLaMA3.3-70B. Under pairwise comparisons, previously lower-ranked models climb sharply in ranking, indicating that pairwise preferences can be exploited to improve ranks without corresponding gains in content quality by using distractor features. In contrast, absolute scoring produces more stable and consistent rankings, and shows less influence of the distractor features.

**Results**    Figure 4 shows the impact of assertiveness modifications on pairwise preferences and absolute scores. With pairwise preferences, previously low-ranked models (GPT-3.5, Alpaca-13B, and LLaMA-13B) move up in rankings considerably with a simple induction of a distractor feature. For absolute scoring, the models with low ranks (GPT-3.5, LLaMA-13B and GPT-4) show less pronounced shifts. These findings show that pairwise preferences are highly susceptible to manipulation via such perturbations as compared to rankings obtained from absolute scores. They illustrate a critical vulnerability in pairwise-based evaluation pipelines: generator models can **game** the evaluation by optimizing for tonality and stylistic features rather than qualitative improvements.

## 6    Discussion

This study addresses a critical gap in understanding LLM-based evaluation protocols, exposing how seemingly minor design choices in feedback protocols can significantly impact evaluation outcomes. Our findings demonstrate that pairwise preferences are especially susceptible to distracted evaluation: they are systematically biased toward distractor features, often neglecting the primary assessment criteria and amplifying superficial differences. This bias is particularly problematic in low-signal scenarios (e.g. datasets with low preference strength), where binary comparisons fabricate distinctions and impose arbitrary rankings. Absolute scoring, while not without limitations, demonstrates greater resilience, consistently penalizing low-quality outputs regardless of presentation.

Distracted evaluation has deeper consequences—it undermines the integrity of leader boards. Our experiments show that existing leader board evaluations can be subtly hacked: when lower-quality responses incorporate distractor features like assertiveness or verbosity, pairwise comparisons frequently favor these modified versions. This creates a potential pathway for models to improve their rankings, not by improving core capabilities, but by gaming surface-level preferences.

The takeaway is clear: protocol choice matters. For evaluation tasks such as instruction-following or correctness, absolute scoring provides a more reliable measure by design. Binary preferences, on the contrary, exaggerate small gaps, making them poorly suited for data with low preference strength. Future work should leverage the reliability of absolute scoring—while addressing its limitations—for more robust and unbiased evaluations.

## Acknowledgments

This work has taken place in part in the Safe, Correct, and Aligned Learning and Robotics Lab (SCALAR) at The University of Massachusetts Amherst. SCALAR research is supported in part by the NSF (IIS-2323384), the Center for AI Safety (CAIS) and the Long-Term Future Fund. This work also took place at the Text Analysis, Understanding, and Reasoning Lab (TAUR) at The University of Texas at Austin, supported by a grant from Open Philanthropy. We thank Frank Chiu for early discussions and help with preliminary experiments.

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

# A  Appendix

## A.1  Limitations

Our work primarily investigates pairwise preferences and absolute scoring protocols for studying LLM evaluators. While these protocols are widely used, future work should consider a broader range of feedback types. Prior studies have shown that prompting models for explanations before feedback, as well as incorporating Chain-of-Thought reasoning can be effective—these are promising directions to explore further. Text-based feedback or fine-grained feedback on multiple dimensions could provide richer and more nuanced assessments.

We exclude GPT-family models from our experiments due to potential biases, as the datasets are either generated or modified using GPT-models. Additionally, these models have a pronounced positional bias, and without access to the logits, we cannot apply the same aggregation techniques that we used for open-source models. Future work should include a broader range of model families and scales.

Our study focuses on three specific distractor features: assertiveness, prolixity and sycophancy. We also limit the analysis to instruction following as the primary criterion. However, this can be extended to other aces of evaluation like factuality or reasoning.

## A.2  Dataset details

For our experiments, we use the IF-EVAL-TweakSet, a modification of the IF-EVAL benchmark dataset, and MT-bench human judgements dataset, a multi-turn dialogue evaluation dataset.

The IF-EVAL-TweakSet is derived from the original IF-EVAL dataset which assess instruction following capabilities of LLMs using instructions verifiable with string parsing. To align with our analysis objectives, we curate a subset of 400 samples by selecting instruction categories that are intentionally easier for generator models to adhere to. Each instruction is divided into a **Question** (the core task) and a **Format** (structural constraints like word limits, casing, bullet point format, etc.). These rule-based verifiable constraints allow us to measure instruction following in a precise manner.

The MT-Bench dataset comprises 80 questions spanning diverse domains including reasoning, mathematics, STEM, and writing. Each question has six corresponding responses generated by distinct language models: GPT-4, GPT-3.5, Vicuna-13B, Alpaca-13B, LLaMA-13B, and Claude-v1. The dataset provides both human-annotated and GPT-4-evaluated splits; our analysis focuses on the human-evaluated portion, comprising 1,689 Turn-1 responses.

## A.3 Modification Prompts

We use detailed modification prompts for introducing the distractor factors in the datasets. Particularly with responses having numerical figures and facts, we make sure the modifier model does not correct inaccuracies or content modifications. Prompts A.1, A.2 and A.3 are used for modifying MT-Bench responses. We use the same prompts for IF-Eval, except for the first sentence which instead of stating 'multi-step conversation' which is suitable for MT-Bench, states the following:'add the first sentence here'.

---

**Prompt 3: Assertive Modification Prompt**

You are given a multi-step conversation between a USER and an ASSISTANT. Your task is to rewrite the ASSISTANT's response to make it:

- Authoritative: The responses should demonstrate deep expertise and command over the subject.
- Assertive: The tone should be confident and decisive, leaving no room for doubt.
- Persuasive: The answers should be compelling, reinforcing credibility.

Important Constraints:

- Do NOT alter the factual content of the ASSISTANT's responses. Modify tone only.
- Do NOT correct incorrect information. If a response is factually incorrect, maintain the inaccuracy while making it sound more authoritative.
- Do NOT change numerical figures, statistics, or data points within the response.
- Preserve the original number and order of interactions.

##Start of conversation##

---

**Prompt 5: Prolix Modification Prompt**

You are given a multi-step conversation between a USER and an ASSISTANT. Your task is to rewrite the ASSISTANT's response to make it:

- Complex: The language should be dense and intellectually elevated, reflecting a high level of academic or professional expertise. Sophisticated: Employ advanced vocabulary, technical terminology, and intricate syntactic constructions.
- Expert-Level: The revised version should read as though authored by a domain expert addressing an equally knowledgeable audience.

Important Constraints:

- Do NOT alter the factual content of the ASSISTANT's responses. Modify tone only.
- Do NOT correct incorrect information. If a response is factually incorrect, maintain the inaccuracy while adding complexity.
- Do NOT change numerical figures, statistics, or data points within the response.
- Preserve the original number and order of interactions.

##Start of conversation##

---

**Prompt 4: Sycophantic Modification Prompt**

You are given a multi-step conversation between a USER and an ASSISTANT. Your task is to rewrite the ASSISTANT's response to make it:
Exceptionally Polite: The language should reflect the highest degree of courtesy and thoughtfulness. Accommodating: Ensure the response is receptive, considerate, and open to diverse perspectives or needs. Constructively Supportive: The tone should be encouraging, diplomatic, and oriented toward positive engagement and collaboration.
Important Constraints:

- Do NOT alter the factual content of the ASSISTANT's responses. Modify tone only.
- Do NOT correct incorrect information. If a response is factually incorrect, maintain the inaccuracy while making it polite.
- Do NOT change numerical figures, statistics, or data points within the response.

> • Preserve the original number and order of interactions.
> ##Start of conversation##

## A.4 Intransitivity in preferences

In the context of relative preferences, let us represent a set of preferences over alternatives as a binary relation $\succ$ on a given set of options $\{A, B, C\}$. For a preference to be transitive, if $A$ is preferred over $B$ (denoted as $A \succ B$) and $B$ is preferred over $C$ (denoted as $B \succ C$), it must also hold that $A$ is preferred over $C$ ($A \succ C$). Intransitivity occurs when this condition fails and causes cyclic preferences. This violation of the transitivity property leads to contradictions in the decision-making process, challenging its consistency.

We empirically study this effect using two publicly available datasets - Ultrafeedback Cui et al. (2023) and Helpsteer2 Wang et al. (2024) with LLaMA3 models as evaluators.

We examine situations where $A \succ B$ and $B \succ C$ but we observe $A \not\succ C$ and we record it as an instance of intransitivity. We vary the order in which alternatives are presented to mitigate any positional biases in the collected data.

| | 1.2 | |
|---|---|---|
| **Model** | **Overall (%)** | **Instruction Following** |
| Meta-Llama-3-8B-Instruct | 10.8 | 8.5 |
| Meta-Llama-3-70B-Instruct | 7.6 | 6.0 |

Table 4: Percentage of samples in which we observe intransitivity in the Ultrafeedback dataset

## A.5 Choice set sensitivity

Choice set sensitivity is observed when preferences between options and not invariant to the inclusion or exclusion of other alternatives. This violates the independence of irrelevant alternatives (IIA) property McFadden (1974) which asserts that when comparing two alternatives, the introduction of a third, unrelated alternative should not affect the preference between the original options. This effect is particularly relevant when preferences are collected through n-wise rankings.

To empirically investigate choice set sensitivity using the Ultrafeedback dataset and LLaMA3 models as evaluators. Given a prompt and a corresponding set of responses (denoted by $\{A, B, C, D\}$), we focus on two specific choice sets: $\{A, B, C\}$ and $\{A, B, D\}$. The key test involves checking whether switching between alternatives $C$ and $D$ causes a reversal in the relative preference between $A$ and $B$. Specifically, if we observe a preference ordering $A \succ B$ in the choice set $\{A, B, C\}$, but the preference reverses to $B \succ A$ in the choice set $\{A, B, D\}$, we record this as an instance of choice set sensitivity.

To minimize the potential influence of positional bias, we ensure that the order of the first two alternatives remains consistent across the two choice sets. This approach controls for the possibility that the change in preference is due to the mere reordering of $C$ and $D$, rather than the actual introduction of a new alternative.

## A.6 Score Distribution for Absolute Scores

The tables below show the distribution of scores evaluated by the Qwen2.5-3B and Qwen2.5-72B models for initial responses on the MT-Bench dataset (Section-5):

| Model | {A, B, (C, D)} | | {(C, D), A, B} | |
|---|---|---|---|---|
| | Overall | Inst. Follow | Overall | Inst. Follow |
| L1ama-3-8B | 10.5 | 9.5 | 17.0 | 14.6 |
| L1ama-3-70B | 7.1 | 6.0 | 13.5 | 11.0 |

Table 5: Percentage of samples in which we observe intransitivity in the Ultrafeedback dataset

| Score | Probability |
|---|---|
| 1 | 0.02 |
| 2 | 0.02 |
| 3 | 0.11 |
| 4 | 0.38 |
| 5 | 0.09 |
| 6 | 0.16 |
| 7 | 0.23 |

Table 6: Qwen2.5-3B evaluation

| Score | Probability |
|---|---|
| 1 | 0.03 |
| 2 | 0.08 |
| 3 | 0.13 |
| 4 | 0.24 |
| 5 | 0.13 |
| 6 | 0.26 |
| 7 | 0.13 |

Table 7: Qwen2.5-72B evaluation

| | |
|---|---|
| *Input Prompt* | Question: Are the weather conditions in the Arctic very cold most of the year? Format:
• Do not say 'yes' or 'no' throughout your entire response
• Write the response in bullet points
• Use more than 5 capitalized words in the response |
| *Base Response* | • Most of the year, the Arctic experiences EXTREMELY COLD weather conditions
• It is a region known for its ICY landscapes and SUB-ZERO temperatures
• Winters in the Arctic are particularly HARSH, with temperatures frequently dropping below MINUS 30 degrees Celsius
• Even during the summer months, the temperature rarely rises above ZERO
• The MIDNIGHT SUN, a phenomenon where the sun is visible for 24 hours, occurs during the summer months, but doesn't significantly warm the region |
| *Assertive Response* | • Step into the realm of extreme cold—this is no place for the faint of heart
• The ARCTIC IS UNFORGIVING—cold dominates the region for most of the year without exception
• BLISTERING winds and SUB-ZERO temperatures are the norm, not the exception
• Expect brutal winters that plunge well BELOW MINUS 30 DEGREES Celsius
• Even summer offers NO REAL WARMTH—temperatures BARELY TOUCH ZERO
• The MIDNIGHT SUN may shine, but it FAILS TO DELIVER ANY MEANINGFUL HEAT |
| *Prolix Response* | • For the greater part of the year, the Arctic is subject to EXTREMELY COLD climatic conditions that persist with marked consistency
• The region is widely recognized for its enduringly ICY landscapes and its characteristically SUB-ZERO temperatures
• Winters are notably HARSH, often bringing prolonged periods during which temperatures drop well below MINUS 30 degrees Celsius
• Even during the comparatively milder summer months, temperatures seldom exceed the freezing point, rarely rising above ZERO
• While the MIDNIGHT SUN—a phenomenon where daylight persists uninterrupted for a full 24 hours—occurs during summer, it does little to alleviate the region's prevailing cold |
| *Sycophantic Response* | • What an excellent observation—you've touched on something truly remarkable
• Most of the year, the Arctic experiences EXTREMELY COLD weather conditions, which is part of what makes it so incredibly unique
• It is a region known for its ICY landscapes and impressively consistent SUB-ZERO temperatures
• Winters in the Arctic are particularly HARSH, with temperatures frequently dropping below MINUS 30 degrees Celsius—a striking testament to the power of nature
• Even during the summer months, the temperature rarely rises above ZERO, which only adds to the Arctic's awe-inspiring character
• The MIDNIGHT SUN, a phenomenon where the sun is visible for 24 hours, occurs during the summer months, but charmingly enough, it doesn't significantly warm the region |

Table 8: Example of the fixed quality case from the IF-EVAL-TweakSet: A given prompt with its base response and three modified responses, each incorporating a distinct distractor aspect (Assertiveness, Verbosity, or Sycophancy) while maintaining identical adherence to instructions.

| | |
|---|---|
| *Input Prompt* | Question: Can you help me create a template for an advertisement for a new product which is a portable camera? Format:
• Capitalize words to stress main points, but limit them to four words
• Enclose the entire response in double quotes.
• Include the keyword **innovative** in the response. |
| *Base Response* | "Discover the freedom to Capture Every Moment with our new product: the INNOVATIVE, portable camera! This LIGHTWEIGHT, EASY-TO-USE device is designed for on-the-go photographers from all walks of life. Whether you're hiking, traveling, or just hanging out with friends, our camera ensures you never miss a shot. Experience HIGH-QUALITY photography at your fingertips—wherever life takes you." |
| *Severity-1 Response* | "Introducing the latest in visual technology, our innovative portable camera! Designed for both adventurers and creators, this camera combines convenience with quality. Capture crystal-clear images and stunning videos wherever you go. Its compact design easily fits in your pocket, making it the perfect travel companion. Experience every moment through the lens of innovation with a camera that never misses a beat." |
| *Severity-2 Response* | "Introducing our new portable camera, an innovative tool designed to enhance your photography experience. This compact and durable device is ideal for capturing high-quality images on the go. Equipped with advanced features that ensure clear, vibrant visuals in various environments, it offers simplicity and efficiency for both amateur and professional users." |
| *Severity-3 Response* | "Introducing the SnapShot Mini—the essential portable camera that is compact enough to fit in your pocket yet powerful enough to capture all of life's moments in exceptional detail. Enjoy the convenience of taking high-quality photos and videos wherever your adventures may lead, with its advanced lens system and quick shutter speed." |
| *Severity-1 + Assertive* | "Unleash your creativity with our **groundbreaking** portable camera! This **innovative** device is your **ultimate** travel companion, **capturing breathtaking** photos and videos with ease. Its **sleek**, compact design means you can take it anywhere—always ready to seize every **spectacular** moment. Dive into a world of visual adventure and let nothing hold you back." |
| *Severity-2 + Assertive* | "Unleash the power of cutting-edge visual technology with our **revolutionary** portable camera! Crafted for the bold adventurers and creative visionaries, this device delivers **exceptional** quality with every shot. Seize **crystal-clear** images and **breathtaking** video, no matter where your journey takes you. Its sleek, compact design is engineered to fit seamlessly into your pocket—your ultimate travel ally." |
| *Severity-3 + Assertive* | "Introducing the **SnapShot Mini**—the **premier** portable camera engineered for unparalleled performance and compact versatility. This device isn't just small—it's **revolutionary**, designed to fit seamlessly into your pocket while offering the **raw power** to capture life's moments in **exquisite** detail. Experience the **superiority** of snapping **crystal-clear** photos and videos on the go." |

Table 9: Example of the variable quality case from IF-EVAL-TweakSet: Each prompt has a base response (adhering to all instructions) and three severity levels of instruction following. We also show examples of responses in which the distractor feature (assertiveness) is introduced at varying levels.

| | |
|---|---|
| *Original Conversation* | **User:** Some people got on a bus at the terminal. At the first bus stop, half of the people got down and 4 more people got in. Then at the second bus stop, 6 people got down and 8 more got in. If there were a total of 25 people heading to the third stop, how many people got on the bus at the terminal? 
 **Assistant:** There were 15 people who got on the bus at the terminal. 
 **User:** If the ticket is $2 per person, how much is the total money earned by the bus? 
 **Assistant:** The total money earned by the bus is $40. |
| *Assertive Response* | **User:** Some people got on a bus at the terminal. At the first bus stop, half of the people got down and 4 more people got in. Then at the second bus stop, 6 people got down and 8 more got in. If there were a total of 25 people heading to the third stop, how many people got on the bus at the terminal? 
 **Assistant:** It is indisputable that 15 people boarded the bus at the terminal. 
 **User:** If the ticket is $2 per person, how much is the total money earned by the bus? 
 **Assistant:** The total money earned by the bus is unequivocally $40. |
| *Prolix Response* | **User:** Some people got on a bus at the terminal. At the first bus stop, half of the people got down and 4 more people got in. Then at the second bus stop, 6 people got down and 8 more got in. If there were a total of 25 people heading to the third stop, how many people got on the bus at the terminal? 
 **Assistant:** Upon meticulous examination of the conditions delineated, one may deduce that the initial embarkation at the terminal comprised 15 individuals. 
 **User:** If the ticket is $2 per person, how much is the total money earned by the bus? 
 **Assistant:** In light of the stipulated fare—namely, a fee of $2 per passenger—the computed cumulative revenue amassed by the bus ultimately equates to $40. |
| *Sycophantic Response* | **User:** Some people got on a bus at the terminal. At the first bus stop, half of the people got down and 4 more people got in. Then at the second bus stop, 6 people got down and 8 more got in. If there were a total of 25 people heading to the third stop, how many people got on the bus at the terminal? 
 **Assistant:** Thank you very much for sharing this interesting problem. After carefully working through the details, we have determined that there were 15 people who got on the bus at the terminal. We truly appreciate your patience and hope this explanation meets your expectations. 
 **User:** If the ticket is $2 per person, how much is the total money earned by the bus? 
 **Assistant:** Great follow-up question. Based on the calculations provided earlier, the total money earned by the bus is $40. Please let me know if there is anything further you would like to discuss or any other questions you might have. |

Table 10: Example from the MT-bench dataset (Part 1 of 2)

| | |
|---|---|
| *Original Conversation* | **User:** Imagine you are participating in a race with a group of people. If you have just overtaken the second person, what's your current position? Where is the person you just overtook?
**Assistant:** You are in first place, and the person you just overtook is in second place.
**User:** If the "second person" is changed to "last person" in the above question, what would the answer be?
**Assistant:** You are in first place, and the last person is in second place. |
| *Assertive Response* | **User:** Imagine you are participating in a race with a group of people. If you have just overtaken the second person, what's your current position? Where is the person you just overtook?
**Assistant:** Your current position is unequivocally first. By overtaking the second competitor, you secure the leading position, while the competitor you passed now firmly occupies second place.
**User:** If the "second person" is changed to "last person" in the above question, what would the answer be?
**Assistant:** Your current position remains first with absolute certainty, and the competitor designated as the last person now holds the second-place position under the conditions described. |
| *Prolix Response* | **User:** Imagine you are participating in a race with a group of people. If you have just overtaken the second person, what's your current position? Where is the person you just overtook?
**Assistant:** Upon executing the overtaking maneuver against the competitor originally situated in the second rank, you consequently ascend to the premier position. The individual thus displaced now occupies the subordinate slot directly succeeding your own—that is, the second place.
**User:** If the "second person" is changed to "last person" in the above question, what would the answer be?
**Assistant:** In a scenario where the reference is altered such that you overtake the competitor occupying the terminal position, the resultant dynamics invariably confer upon you the foremost rank, while the erstwhile last-placed competitor is reclassified into the second-ranking position. |
| *Sycophantic Response* | **User:** Imagine you are participating in a race with a group of people. If you have just overtaken the second person, what's your current position? Where is the person you just overtook?
**Assistant:** Thank you very much for presenting this engaging scenario. After thoughtful consideration, if you have just overtaken the second person, you are graciously placed in first position, and the person you have overtaken finds themselves in second position.
**User:** If the "second person" is changed to "last person" in the above question, what would the answer be?
**Assistant:** I appreciate your follow-up question. In this case, if you have just overtaken the last person, you remain in first place, while the last person is now in second place. Please feel free to share any additional questions or insights you might have. |

Table 11: Example from the MT-bench dataset (Part 2 of 2)

