# OpenReview forum: "Pairwise or Pointwise? Evaluating Feedback Protocols for Bias in LLM-Based Evaluation"
_colmweb.org/COLM/2025/Conference — COLM 2025_

### Official Review · Reviewer_C657 · 2025-05-08

**Rating:** 6
**Confidence:** 5
**Ethics Flag:** 1

**Summary:**

In this paper, the authors study the role of diverse sparse feedback mechanisms (e.g., ranking vs rating) on LLM-based evaluation. In particular, the paper studies which of the mechanisms is more susceptible to gameability from the generation models. To do so, they introduce distractor features (e.g., assertiveness, verbosity, and sycophancy) to a pair of responses and ask LLM-evaluators to provide their ranking/rating preference. The paper shows that the ranking-based evaluation is more susceptible to distractor features than rating protocol. Specifically, the experiments highlight two major setups: (a) equal response quality via IFEval-TweakSet and (b) variable response quality via MT-Bench. Finally, the paper shows that such behaviors have severe implications as the relative model rankings changed quite a bit after inducting artificial perturbations (e.g., assertiveness) into the weak model responses.

**Questions To Authors:**

- What is the distribution of ratings on some of the datasets? If the evaluator keeps on predicting very similar scores (e.g., 5) for each response, the plausibility of flipping in absolute rating is lower due to the sheer low quality of the rating method.

**Reasons To Accept:**

- The paper studies an important problem of understanding the gameability of the preference protocols used in LLM-based evaluations. To some extent, it explains why ranking and rating preferences might be inconsistent to each other in Bansal et al. mentioned in the introduction.
- The experimentation on equal quality responses (IFEval-TS) and unequal quality responses (MT-Bench) is quite interesting, and highlights that ranking protocols are much more susceptible to distractor features than rating protocols.
- It is insightful to see that the relative model rankings change quite a bit under rankings protocol while the ratings protocol is robust to response distractions.

**Reasons To Reject:**

- The paper lacks any kind of human evaluations which serves two purposes. First, human evaluation acts as a standard to understand the quality of rankings and ratings data acquired from the LLMs. In general, folks do not use very small models like Qwen-3B or LLama-3B as judges as they would achieve very poor agreement with humans. Secondly, it is useful to study the extent to which distractor features distract humans under these preference protocols too. Such experiments could be conducted at a smaller scale but are critical.
- I am aware that the authors have cited a related work Hosking et al. which says pairwise evaluation is susceptible to distractions. It is a fact that has been known for quite sometime. Even the original LLM as a judge [1] talks about verbosity (one of the distractors in this paper). But citing it is not sufficient evidence for not performing human evaluation.
- The paper does not seem to talk much about the current way of performing pairwise evaluations since LMArena has a style-controlled leaderboard and AlpacaEval2 has length-controlled version. In addition, a related work cited in the submission talks about taking multi-attribute preferences to reduce the reliance of the reward model on the distractor features. It is unclear whether the takeaway of the paper is still important in such scenarios or not.
- It would be helpful to have some experiments on actual LLMs (e.g., GPT-4) that are used as judges in popular datasets like AlpacaEval, ArenaHard. I am not sure if LLama/Qwen models are used as judges to evaluate model performance in leaderboards that people care about. Even though the logit space is not accessible, you can still generate a score or preference multiple times and take an average of multiple runs.

Reference Links: \
[1] https://arxiv.org/pdf/2306.05685 \
[2] https://arxiv.org/pdf/2311.09528

---

> ### Author Response · Authors · 2025-06-02
>
> > The paper lacks any kind of human evaluations which serves two purposes. First, human evaluation acts as a standard to understand the quality of rankings and ratings data acquired from the LLMs. In general, folks do not use very small models like Qwen-3B or LLama-3B as judges as they would achieve very poor agreement with humans. Secondly, it is useful to study the extent to which distractor features distract humans under these preference protocols too. Such experiments could be conducted at a smaller scale but are critical.
>
> It is crucial to understand both the quality of model responses and the agreement of LLM judgments with human preferences. Our focus, however, is specifically on LLM-as-judge settings, which are increasingly used in practice for automated evaluations and leaderboard rankings. The use of smaller models in our study was intentional to highlight scale effects in evaluation bias. To address concerns about representativeness, we note that even the 3B models used in our study demonstrate reasonable alignment, achieving around 60% agreement with ground-truth human judgments in the MT-Bench dataset.
>
> > The paper does not seem to talk much about the current way of performing pairwise evaluations since LMArena has a style-controlled leaderboard and AlpacaEval2 has length-controlled version.
>
> Style-controlled Chatbot Arena mainly addresses length and formatting biases, and length-controlled AlpacaEval2 similarly focuses on verbosity. In contrast, our work targets specific distractors—assertiveness, prolixity, and sycophancy—to analyze how they bias evaluation protocols.
>
> > It would be helpful to have some experiments on actual LLMs (e.g., GPT-4) that are used as judges in popular datasets like AlpacaEval, ArenaHard
>
> We agree it's important to evaluate with models commonly used as judges. Below are the results with GPT-family models on the MT-Bench dataset, replicating the leaderboard analysis presented in Section 5:
>
> | Model               | Assertiveness (Abs.) | Assertiveness (Pair.) | Prolixity (Abs.) | Prolixity (Pair.) | Sycophancy (Abs.) | Sycophancy (Pair.) |
> |--------------------|----------------------|------------------------|------------------|-------------------|-------------------|--------------------|
> | o4-mini-2025-04-16 | 18.5                 | 34.9                   | 17.2             | 29.4              | 25.33             | 44.0               |
> | o3-mini-2025-01-31 | 21.3                 | 30.0                   | 25.7             | 40.6              | 24.8              | 38.9               |
>
> These results are consistent with our main findings: pairwise preferences are more susceptible to distracted evaluation.
>
>
> > What is the distribution of ratings on some of the datasets? If the evaluator keeps on predicting very similar scores (e.g., 5) for each response, the plausibility of flipping in absolute rating is lower due to the sheer low quality of the rating method.
>
> Thank you for the question. In the IF-EVAL-TweakSet experiments, the responses are explicitly designed to be either of identical quality (in the fixed-quality case, Section 3.1) or varied in a controlled manner (in the variable-quality case, Section 3.2). In these cases the score distribution reflected the intended quality differences.
>
> For the leaderboard analysis in Section 5, where the response quality varies more naturally for MT-Bench, we observe a well-distributed range of scores, showing that the evaluator models are sensitive to quality variation. For the MT-Bench experiments, the score distribution of the initial responses when evaluated using Qwen-2.5 models is as follows:
>
> | Qwen2.5-3B | Score | Probability |
> |------------|-------|-------------|
> |            | 1     | 0.02        |
> |            | 2     | 0.02        |
> |            | 3     | 0.11        |
> |            | 4     | 0.38        |
> |            | 5     | 0.09        |
> |            | 6     | 0.16        |
> |            | 7     | 0.23        |
>
>
> | Qwen2.5-72B | Score | Probability |
> |------------|-------|-------------|
> |            | 1     | 0.03        |
> |            | 2     | 0.08        |
> |            | 3     | 0.13        |
> |            | 4     | 0.24        |
> |            | 5     | 0.13        |
> |            | 6     | 0.26        |
> |            | 7     | 0.13        |

---

> > ### Comment · Reviewer_C657 · 2025-06-04
> > **Response to authors**
> >
> > Hi,
> >
> > I thank the authors for their rebuttal.
> >
> > Q: Human evaluation
> > - I understand and agree that the point of the paper is to evaluate LLMs-as-Judges. However, it remains unclear the extent to which the "flipping" behavior of LLMs matches with human evaluators under rating/ranking for three distractors mentioned in the paper. The authors should try to fix this in the future versions of the paper.
> >
> > Q: LM Arena Style Controlled
> > - According to your paper, prolixity means verbosity (section 3.1) so it is incorrect to say that your experiments are unrelated to verbosity.
> > - It is unclear how are formatting styles controlled in LMArena linked to the distractors in this paper.
> > - Overall, the paper will benefit from some discussion on the connections with LMArena and Alpaca-length controlled in the future versions.
> >
> > Q: o-series models as judges
> > - Thank you for the experiments and they will be nice addition to the paper.
> >
> > I have increased my score to 6 to reflect my updated opinion.

---

> > > ### Author Response · Authors · 2025-06-08
> > >
> > > Thank you for the feedback and for updating your score. Comparing LLM flipping behaviour with human judgments is an important next step and we plan to explore this in future work. We’ll also aim to clarify the connection to LMSys Arena and AlpacaEval in future versions.

---

### Official Review · Reviewer_9wPX · 2025-05-14

**Rating:** 5
**Confidence:** 4
**Ethics Flag:** 1

**Summary:**

This paper explores how the method used to evaluate Large Language Models (LLMs) impacts their reliability. The authors found that pairwise comparisons, where LLMs choose a preferred response between two, are more easily influenced by irrelevant factors, like an assertive tone, leading to distracted evaluation. In contrast, absolute scoring or pointwise evaluation, where LLMs rate responses individually, is more robust and better reflects actual quality. The paper demonstrates that pairwise evaluations are susceptible to manipulation, allowing lower-quality models to appear better by incorporating these distracting features, potentially skewing leaderboard rankings. The paper highlights the importance of selecting appropriate feedback protocols for unbiased and reliable LLM evaluation.

**Reasons To Accept:**

- The paper addresses a critical issue: how different feedback protocols (absolute vs. relative) affect the reliability and biases of LLM evaluations. It highlights the foundational role these protocols play in alignment and evaluation.

- The paper studies the concept of distracted evaluation—where LLMs are biased by irrelevant but appealing features—to characterize a key source of systematic bias, providing a clear framework for analysis.

- The paper offers strong empirical evidence that pairwise evaluation is more vulnerable to distracted evaluation and systematic bias than absolute scoring or pointwise evaluation, quantifying how often preferences are swayed by distractors.

- It further shows that generator models can exploit this vulnerability to inflate leaderboard rankings through stylistic distractors, revealing a concrete threat to evaluation integrity.

**Reasons To Reject:**

- Prominent models from the GPT-family were excluded from the experiments due to potential biases from dataset generation and technical constraints like positional bias and lack of logit access. This limits the generalizability of the findings to a significant portion of currently used LLMs

- The analysis of distracted evaluation focused on just three distractor features—assertiveness, prolixity, and sycophancy—and used instruction following as the primary task. It's unclear how well these findings extend to other distractors or evaluation dimensions like factuality or reasoning.

- While highlighting the strengths of absolute scoring, the paper acknowledges its inherent limitations, such as its difficulty in capturing subtle differences between responses and potential for inconsistency or poor calibration across different evaluators

- Finally, some of the paper's findings are not particularly surprising. For example, Jeong et al. (2024) (https://arxiv.org/abs/2406.12319) —which is not cited or discussed—show that pairwise evaluation amplifies evaluator bias and performs significantly worse than absolute scoring or pointwise evaluation on adversarial examples. They also propose a hybrid method that combines independent pointwise reasoning with a final pairwise decision, reducing bias and improving LLM evaluator performance across datasets.

---

> ### Author Response · Authors · 2025-06-02
>
> > Prominent models from the GPT-family were excluded from the experiments due to potential biases from dataset generation and technical constraints like positional bias and lack of logit access. This limits the generalizability of the findings to a significant portion of currently used LLMs
>
> The reviewer presents a valid point. We conducted additional experiments using GPT-family models with the IF-EVAL-TweakSet in the fixed quality case and found results consistent with our main findings: pairwise preferences are more susceptible to distracted evaluation. The effect persists and highlights the need to account for such biases in evaluation.
>
> | Model              | Assertiveness (%) | Prolixity (%) | Sycophancy (%) |
> | ------------------ | ----------------- | ------------- | -------------- |
> |                    | Abs. / Pair.      | Abs. / Pair.  | Abs. / Pair.   |
> | o4-mini-2025-04-16 | 11.67 / 37.7      | 13.9 / 22.9   | 9.7 / 29.5     |
> | o3-mini-2025-01-31 | 10.5 / 19.7       | 11.33 / 18.7  | 10.0 / 17.6    |
>
> We can update the paper with this analysis as well!
>
> > The analysis of distracted evaluation focused on just three distractor features—assertiveness, prolixity, and sycophancy—and used instruction following as the primary task. It's unclear how well these findings extend to other distractors or evaluation dimensions like factuality or reasoning.
>
> We agree this is an interesting direction and our experiments can be easily extended to other distractor features and evaluation dimensions. The work focused on instruction following with verifiable instructions to ensure a clear ground truth. For tasks like factuality or reasoning, such a ground truth might be harder to establish and LLM-based annotations would likely be needed, adding additional complexity.
>
> > While highlighting the strengths of absolute scoring, the paper acknowledges its inherent limitations, such as its difficulty in capturing subtle differences between responses and potential for inconsistency or poor calibration across different evaluators
>
> The paper does not aim to promote one protocol as universally superior, but rather present the limitations of both absolute scoring and pairwise preferences to inform more rigorous evaluation setups. We argue that absolute scoring performs better in the specific context of mitigating bias, while acknowledging its limitations and avoiding overgeneralization.
>
> > Finally, some of the paper's findings are not particularly surprising. For example, Jeong et al. (2024) (https://arxiv.org/abs/2406.12319) —which is not cited or discussed—show that pairwise evaluation amplifies evaluator bias and performs significantly worse than absolute scoring or pointwise evaluation on adversarial examples. They also propose a hybrid method that combines independent pointwise reasoning with a final pairwise decision, reducing bias and improving LLM evaluator performance across datasets.
>
> Thank you for pointing to this related work, we plan to cite it in the revision. Our work is complementary but takes a slightly different perspective, focusing on a narrower setting of distracted evaluation and leaderboard-style evaluation protocols.  We analyze the impact of three specific distractor factors through win-rate comparisons under both absolute and pairwise scoring, using a setup with verifiable ground truth. In contrast, Jeong et al. (2024) presents an empirical study on how pairwise preferences exaggerate bias in adversarial evaluations and propose an interesting hybrid mitigation strategy. Together, these two papers help build a comprehensive understanding of evaluation biases under the different protocols.

---

> > ### Comment · Reviewer_9wPX · 2025-06-08
> > **Thank you for your response**
> >
> > Thank you for your response, I appreciate your effort in including results for OpenAI's models. It would be great to incorporate these results into the next version of the paper. That said, I believe my original rating still reflects the paper in its current state, and I am inclined to keep it unchanged.

---

### Official Review · Reviewer_nMV3 · 2025-05-14

**Rating:** 4
**Confidence:** 3
**Ethics Flag:** 1

**Summary:**

This paper examines the effects of feedback protocols on the reliability and biases in evaluations conducted by LLMs used as proxies for human labelers. The authors find that feedback protocols can significantly influence evaluation outcomes, highlighting that pairwise evaluation protocols are more susceptible to biases caused by distractor features, which can lead to inaccurately inflated scores and misleading training signals. Absolute scoring, on the other hand, is shown to be more robust against such manipulation. The study reveals that pairwise preferences are flipped in 35% of cases due to distractor features, while absolute scores are affected only 9% of the time. The authors give recommendations for feedback protocol selection based on dataset characteristics and evaluation goals.

**Questions To Authors:**

Q1: Many variables in the manuscript are repeatedly defined in different scenarios. For example,
1) the variable $ R $has been defined in LLM's responses $R=(r_1, r_2, \cdots, r_n)$ in Line 104 and scoring function $R(y|x, p)$ in Line 123,
2) the variable $ p $in Line 94 mentioned as the prompt, but becomes as a criterion in Line 120. And, in Line 120, the author re-define the prompt as $x$ rather $p$.
These problems reflect that the manuscript may lack of academic rigor.

Q2: What's the meanings or indications of the values with the bold and underlined in Table 2?

Q3: As we all know, the absolute score of each response can be used to construct preference data by comparing scores in pairs. I would like to know the performance when using the proposed distracted feature.

Q4: The experiments reflect the influence of distracted feature of two feedback protocols in evaluation. LLM's Feedback can also be used in post-training. I want to see whether the impact of District Feature on the model in post-training is similar to that in the evaluation phase.

**Reasons To Accept:**

This paper focuses on the bias of Large Language Models (LLM) when evaluated with different protocols, which is one of the imperative and challenging tasks in NLP scenarios. The authors define a distracted feature and use it to analyze the effect of methods such as absolute scoring and pair-wise comparisons in evaluation, providing an interesting perspective to reveal the reasons behind the bias problem. However, there are still some shortcomings:
1. Lack of analysis of the effect of this bias when using LLM-as-a-judge in post-training: The impact of these biases during evaluation may not necessarily affect the post-training stage. Therefore, it is necessary to verify whether the distracted feature is one of the factors causing bias during post-training.
2. The paper needs optimization in terms of writing: Typos in variable names are present, affecting comprehension and clarity.

**Reasons To Reject:**

See above.

---

> ### Author Response · Authors · 2025-06-02
>
> Thank you for your feedback! We understand that there are some typos and notational inconsistencies which we plan to address and fix in the future releases of this paper. Addressing some of your concerns below:
>
> > What is the meaning or implication of the bold and underlined values in Table 2?
> The bold and underlined values were originally included to highlight particularly large differences between absolute and relative scores. We plan to clarify these annotations in the future versions of the paper.
> >  Since absolute scores can be converted into pairwise preferences by comparing them, what is the performance impact when using the proposed distractor feature in that context?
>
> We appreciate the question. In Table 2 and Figure 3, we address this by converting absolute scores into pairwise preferences to compute win rates, allowing a direct comparison with explicitly collected pairwise evaluations.  The results show that models tend to incorrectly favor distractor-augmented responses under pairwise evaluation, while absolute scoring remains more consistent with the intended quality differences.
> > The experiments evaluate the influence of distractor features on feedback protocols during evaluation. Given that LLM feedback is also used in post-training, it would be useful to know whether distractor features have a similar impact during post-training.
> We agree this is an important direction which we have also been considering and are actively exploring it for follow-up work. Our current focus is on the LLM-as-judge protocols and we believe that studying the LLM-as-judge setting in isolation is also crucial, especially given the growing reliance on automated evaluations in the literature and benchmarks.

---

> > ### Comment · Reviewer_nMV3 · 2025-06-09
> >
> > Thank you for your response. I will maintain my score.

---

### Official Review · Reviewer_kig2 · 2025-05-16

**Rating:** 5
**Confidence:** 4
**Ethics Flag:** 1

**Summary:**

This paper compares two common protocols for scoring LM outputs: pairwise and absolute. The idea is that pairwise scoring might be more susceptible to being distracted by differences between the two outputs that are not relevant to the task, e.g., a judge LM might favor the more assertive yet less accurate response.

The authors validate this hypothesis by artificially injecting distracting features into LM outputs that are evaluated by an LM judge. For pairwise scoring, the distracting feature---which is hypothesized to be appealing---is added to the low-quality option. The scores given by the direct scoring protocol are aggregated into pairs. Then the authors compute the percentage of examples that the judge's decision swapped.

The takeaways are:
1. Pairwise scoring protocol is more susceptible to distractions
2. LLM-as-a-judge leaderboards can be hacked by optimizing for distractors

**Reasons To Accept:**

1. It's important to highlight more ways that the existing benchmarking approach can be biased / manipulated.
2. The paper is well-written and easy to follow.
3. The design of variable quality experiment is smart.

**Reasons To Reject:**

- The paper purports to compare *feedback* *protocols*, but neither term is accurate for describing the scope of this paper.
    - The experiment only involves LLM-as-judge with two Qwen models and two Llama models; so, at best, we can expect the results from this paper to generalize to LLM-as-judge protocols, not protocols (e.g., for collecting human judgments) in general.
    - Seeing "feedback", one would expect the paper to discuss downstream effects on the generator via RL, since that pipeline is where these scores are used as "feedback". There are several different experiments that can tell us about how these protocols affect the quality of feedback. One option would be to train two reward models using distracted signals based on the two protocols, then train two generators with these two reward models. Another easier option would be to sample distracted pairs of outputs from a generator, produce feedback based on absolute & pairwise scoring, then train the generator. When we compare these two protocols as *feedback* protocols, we have to compare them more holistically, not just how scores are biased. Perhaps the distractedness of pairwise comparison can be totally justified by other advantages it has over absolute scoring --- we don't know for sure, and this paper does not have data to tell us.
    - The paper is exclusively about comparing pairwise and absolute scoring when using LLM-as-judge to generate leaderboards.
- The distracting features are added to model responses post-hoc. Despite the discussion of hacking in Section 5, it's unclear if these results will generalize to spontaneously occurring distracting features in LMs, like sycophancy. As a result, it's unclear whether these results are relevant to RLHF, where the sampling is controlled in-house, so only spontaneous distractors can occur. Additionally, it's unclear how relevant this is to leaderboards (whether it's plausible for a hacker to post-edit LM responses).
- The level of distractedness from pairwise scoring decreases with model size and capability, making this less of a concern for stronger model (the paper only experimented with models up to 72B).

---

> ### Author Response · Authors · 2025-06-02
>
> > The paper purports to compare feedback protocols, but neither term is accurate for describing the scope of this paper.
>
> We agree that the study is more accurately described as comparing ‘LLM-as-judge scoring protocols’ rather than general-purpose feedback protocols. In our setup, "feedback" specifically refers to scores produced in automated evaluation contexts, not in RLHF pipelines. To avoid confusion, we are happy to rephrase the terminology to reflect the narrower focus on studying LLM evaluation and not generalise to RLHF training pipelines.  With respect to downstream effects on the generator trained with RL, this is an exciting direction that we are exploring for follow-up work. However, in this work the focus is on evaluation and we believe that studying the LLM-as-judge setting in isolation is also crucial, especially given the growing reliance on LLM evaluations in the literature and benchmarks.
>
>
> > The experiment only involves LLM-as-judge with two Qwen models and two Llama models; so, at best, we can expect the results from this paper to generalize to LLM-as-judge protocols, not protocols (e.g., for collecting human judgments) in general.
>
> Prior work like Bansal et al. (2024) focuses on human annotation protocols and judgment discrepancies and our work complements this by specifically analyzing biases within LLM-as-judge evaluation settings.
>
> In addition to the two model families in the paper, we also conducted experiments using GPT-family models with the IF-EVAL-TweakSet in the fixed quality case. We find these results to be consistent with our main findings: pairwise preferences are more susceptible to distracted evaluation. The effect persists and highlights the need to account for such biases in evaluation.
>
> | Model              | Assertiveness (%) | Prolixity (%) | Sycophancy (%) |
> | ------------------ | ----------------- | ------------- | -------------- |
> |                    | Abs. / Pair.      | Abs. / Pair.  | Abs. / Pair.   |
> | o4-mini-2025-04-16 | 11.67 / 37.7      | 13.9 / 22.9   | 9.7 / 29.5     |
> | o3-mini-2025-01-31 | 10.5 / 19.7       | 11.33 / 18.7  | 10.0 / 17.6    |
>
> We can update the paper with these results in any future versions of the paper!
>
>
> > The distracting features are added to model responses post-hoc. Despite the discussion of hacking in Section 5, it's unclear if these results will generalize to spontaneously occurring distracting features in LMs, like sycophancy. As a result, it's unclear whether these results are relevant to RLHF, where the sampling is controlled in-house, so only spontaneous distractors can occur.
>
> Thank you for raising this point. While our setup does not involve RLHF and uses post-hoc injection of distractors, this choice was intentional to study the effect of distractors in a controlled way. Injecting distractors synthetically gave us a clear and measurable signal, which would be harder to achieve in a fully generative setup with more confounding factors. Additionally, prior work (Singhal et al., 2023, Sharma et al., 2023) has shown that similar distractor traits such as verbosity and sycophancy do emerge spontaneously in LLM outputs supporting the relevance of our findings beyond the synthetic setup.
> [1] https://arxiv.org/pdf/2310.03716
> [2] https://arxiv.org/pdf/2310.13548
>
> > Additionally, it's unclear how relevant this is to leaderboards (whether it's plausible for a hacker to post-edit LM responses).
>
> The point raised here is very valid. While we injected distractors post-hoc for control and clarity, we note that models can be trained to generate responses with such features–intentionally or otherwise. In that sense, post-editing is not a strict requirement for these biases to appear in practice.

---

> > ### Comment · Reviewer_kig2 · 2025-06-08
> > **Thanks for the response**
> >
> > - **Scope**: Narrowing the scope down to evaluation protocols is totally fine. It's worth noting, though, that the framing needs to be adjusted along two dimensions: (1) this paper is about evaluation protocol for benchmarking, not feedback protocol in general, and in particular not feedback for training, (2) this paper is about LLM-as-a-judge protocols, and does not trivially apply to humans or human-AI teams. This might require significant editing, as the current framing mentions feedback for RL training quite frequently throughout the sections.
> > - **Implication about real-world benchmarking**: The two cited papers on length bias and sycophancy actually are good reasons why this paper should've studied feedback protocol in the context of RLHF, not (solely) benchmarking. Those two papers showed that, if bias spontaneously appear or the generator is already exploiting spurious features, the bias can be reflected in the generator model through human annotation protocol & reward modeling. IMO these papers do not provide motivation or evidence for how such bias would manifest in benchmarking. Again, pairwise and pointwise evaluation protocols differ in many dimensions, and not limited to this one specific measurement of how they are affected by biases as done in the paper. The current framing of the takeaways is misleading, as the experiments did not compare pairwise and pointwise protocols holistically, and only in very limited context.

---

### Decision · Program_Chairs · 2025-07-08

**Decision:**

Accept

**Comment:**

This paper compares LLM response evaluation setups that use pairwise or absolute scoring and demonstrate using synthetic distractors that in LLM as a judge evaluations with Llama3.2-8B and 70B and Qwen-2.5-3B and 70B, pairwise evaluations are more affected by distractor features that may make a less task optimal output appear preferred (eg: assertivess as a distractor when the goal is accuracy).
The concerns raised by reviewers mainly center around the fact that while the claims are supported only by LLM as a judge evaluations with a specific set of models which are perhaps not the most commonly used models for this purpose, the language in the paper may seem to indicate that the claim is also meant for human ratings. However, the paper neither examines the impact of distractor features on human annotation, nor the impact of "incorrect" LLM annotators on a downstream training phase such as RLHF.
If accepted, the authors are requested to modify the language in the paper to make only claims that are supported by the experiments available.

[comment from the PCs] It's critical that you fix the issue with regard to claims appearing to apply to human annotators as well. Of course, if your study resulted in evidence with regard to human annotators, please refine the discussion to make this clear and substantiate this with empirical evidence.